# TokenCount: A Training-Free Framework for Object Counting by Interpreting Output Tokens

## Abstract

Object counting is a critical computer vision task with widespread applications in manufacturing, traffic monitoring, and crowd analysis. Recent class-agnostic object counting methods leveraging the Segment Anything Model (SAM) are limited by the inherent uncertainty of the similarity metric derived from its image encoder. While solutions incorporating additional encoders can refine this similarity, they face challenges due to high computational costs. To overcome this challenge, we propose a novel framework that consists of two critical components working in synergy with SAM. We propose a probabilistic prompt generation stage and an output token-based verification stage. The probabilistic prompt generation stage efficiently generates prompts based on probability distributions from SAM's image embedding, while the output token-based verification stage uses SAM's output tokens to effectively distinguish between positive and negative instances. Experimental results show our method achieves superior accuracy with an MAE of 16.25, outperforming existing training-based and training-free counting methods. Notably, our method achieves comparable performance to training-free approaches that require additional models alongside foundation models. Particularly, on the CARPK dataset, our method achieves superior performance, outperforming all supervised methods and demonstrating comparable results against training-free counting methods. Furthermore, ablation studies prove that this performance gain is critically attributed to two key components. This study not only presents an effective solution for object counting but also showcases the potential of applying foundation models to downstream tasks without fine-tuning and additional models.

## 1 Introduction

Object counting, a fundamental task in computer vision, plays a pivotal role in a wide range of applications, from industrial quality control to urban planning(Zavrtanik et al., 2020; Arteta et al., 2016; Cheng et al., 2019). With the advancement of deep learning, early methods primarily centered on density map estimation and object detection (Ranjan et al., 2021; Yang et al., 2021; Shi et al., 2022; Lu et al., 2019; You et al., 2023; Nguyen et al., 2022; Đukić et al., 2023; Lin & Chan, 2024). However, density maps fail to provide instance-level localization or shape information, while object detection, which approximates objects with bounding boxes, is limited in its ability to precisely represent non-rigid or complex shapes.

Furthermore, a fundamental limitation of these methods is their dependency on large-scale, category-specific datasets for training. To address this, class-agnostic, exemplar-based counting methods were developed(Ranjan et al., 2021; Yang et al., 2021; Shi et al., 2022; Lu et al., 2019; You et al., 2023; Nguyen et al., 2022; Đukić et al., 2023; Lin & Chan, 2024; Huang et al., 2024; Ting et al., 2024; Mondal et al., 2024; Ma et al., 2023; Shi et al., 2024; Kirillov et al., 2023). This method matches features between a user-provided exemplar and a target image, offering the flexibility to count objects of arbitrary categories without task-specific training.

Despite this flexibility, similarity-based methods often struggle to distinguish target objects from visually similar background clutter or distractors. More importantly, these methods still require

large annotated datasets to learn a generalizable embedding space for similarity comparison, thus failing to resolve the bottleneck of data collection.

To overcome these challenges, recent research has shifted towards training-free methods that leverage instance segmentation foundation models. Instance segmentation provides pixel-level masks, enabling precise analysis of objects with complex geometries. Furthermore, foundation models, pre-trained on vast datasets, possess powerful zero-shot generalization capabilities and the ability to extract rich semantic features for diverse categories without fine-tuning.

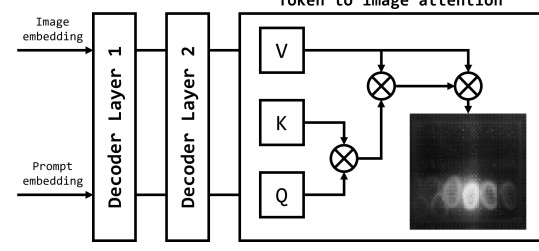

Figure 1: Visualization of attention maps for output token.

However, existing methods based on foundation models still rely on calculating similarity between image embeddings extracted from the model encoder to identify objects. TFOC(Shi et al., 2024), an early work in this area, used embedding similarity as its criterion, but the ambiguity of the similarity itself introduced significant uncertainty. Subsequent studies also focused on counting objects based on vectors derived from additional operations between masks predicted from the model decoder and image embeddings. This approach is inefficient, as the computational overhead scales linearly with the number of prompts.

In response, we propose a novel framework that fully utilizes the intrinsic architecture of foundation models, specifically SAM. Our method consists of two core components:

**Output Token-based Verification.** We directly leverage the output tokens from SAM's decoder. As illustrated in Figure 1, each output token encodes a condensed representation by integrating semantic and positional information through bidirectional attention with the image embedding. This representation activates regions that are semantically and spatially aligned with the input prompt. By using these tokens directly for criterion, we can bypass the uncertainty of similarity and costly post-processing steps of prior methods.

**Probabilistic Prompt Generation.** This stage addresses the limitations of deterministic prompting methods, such as grids or superpixels. Due to the localized nature of output tokens, deterministic approaches often fail to detect objects spatially distant from the initial exemplar. Our proposed method performs a probabilistic exploration that iteratively expands from high-confidence regions, efficiently exploring the entire image while minimizing redundant prompt generation in background areas. The specifics of each stage are detailed in Sections 3.1 and 3.2.

We extensively evaluated our proposed method on the FSC-147 and CARPK benchmarks. The results show that our method outperforms mostly existing training-free methods on both datasets. Notably, on CARPK, it achieves results that are competitive with, and in some cases superior to, supervised methods. Furthermore, ablation studies demonstrate the significant contributions of both the probabilistic prompt generation and the output token-based verification stages to the overall accuracy.

Our contributions are threefold: First, we propose a novel framework that directly repurposes SAM for object counting without architectural modifications or auxiliary models. Second, we propose a probabilistic prompt generation stage that enhances computational efficiency by intelligently exploring the image and reducing redundant background sampling. Third, we propose a output token-based verification stage that utilizes SAM's internal output tokens to accurately discern positive instances without relying on additional post-processing.

## 2 RELATED WORK

### 2.1 CLASS-AGNOSTIC OBJECT COUNTING

Since the introduction of the FSC-147 dataset, class-agnostic object counting methods have studied on training-based methods that analyze correlations between query images and exemplars for ob-

ject counting(Lu et al., 2019). Methods such as FamNet, CFOCNet, Counting-DETR, and LOCA improved accuracy through advanced feature matching, object detection, and adaptive mechanisms, but they shared common limitations of exemplar dependency and high training costs(Ranjan et al., 2021; Yang et al., 2021; Nguyen et al., 2022; Đukić et al., 2023).

Subsequently, training-free methods emerged, leveraging foundation models like SAM, CLIP, or DINO to generate similarity from image embeddings without additional training(Shi et al., 2024; Ting et al., 2024; Lin et al., 2025). Although OmniCount attempted to enhance performance by combining semantic and geometric information, these approaches exhibited performance limitations in the downstream counting task(Mondal et al., 2024).

To address the performance gap with training-based methods, methods like PseCo were proposed, which retain the core architecture of a foundation model while training only a few additional modules(Huang et al., 2024). PseCo achieves high performance by efficiently leveraging the powerful representations of the foundation model, training only lightweight decoder and classification modules to process image and text embeddings.

## 2.2 PROMPT GENERATION IN OBJECT COUNTING

Object counting has recently seen a surge in prompt-based methods leveraging powerful foundation models like SAM and CLIP(Kirillov et al., 2023; Radford et al., 2021). Prompt generation techniques can be broadly categorized into training-free and training-based approaches.

**Training-free Prompt Generation.** A fundamental approach is uniform grid prompting, which places points at regular intervals across an image(Kirillov et al., 2023; Ma et al., 2023). However, this method often misses small objects, wastes computation on background regions, and performs poorly in dense scenes. To better preserve object boundaries, superpixel-based methods like SLIC cluster pixels to generate prompts(Achanta et al., 2012). Nevertheless, their effectiveness is highly dependent on the quality of the superpixel segmentation and can degrade in complex or low-light scenes.

**Training-based Prompt Generation.** PseCo(Huang et al., 2024) employs a trainable point decoder, which is optimized on a heatmap derived from ground truth and initial predictions, to generate effective prompts by extracting the top-$K$ keypoints. (Lin & Chan, 2024) proposed a self-reflective method that uses the predicted density map as a prompt mask for the next iteration. This process is guided by a fixed-point loss function, which iteratively refines low-quality initial prompts until they converge to a stable, high-quality state.

## 3 PROPOSED METHOD

This section explains the two key components that we propose to work in synergy with SAM. First, we introduce a probabilistic prompt generation stage that uses image embedding generated by SAM's image encoder to calculate probability distributions, which are then utilized to efficiently generate prompts. Unlike simple grid or superpixel-based methods, this enables adaptive and efficient placement. Second, we introduce a output token-based verification stage that uses output tokens generated by SAM's decoder to effectively distinguish between positive and negative instances. This is an efficient verification method that doesn't require additional computation, unlike similarity calculation methods through image embeddings proposed by existing studies. Figure 2 shows an overview of our proposed method. It illustrates the entire process that uses the SAM architecture as is, with only our proposed steps added for counting. For illustrative purposes, the distribution diagram below depicts how sampled output tokens are verified as positive or negative at each iteration through the output token-based verification stage.

## 3.1 PROBABILISTIC PROMPT GENERATION

To enhance prompt generation, we propose a probabilistic stage that models the likelihood of object locations across an image. The process begins by computing the cosine similarity between the exemplar mask embedding and the query image embedding, an approach inspired by SAM. We then apply a ReLU function to the similarity map. This operation discards anti-correlated features by clipping all negative similarity values to zero. This effectively treats any region that is not posi-

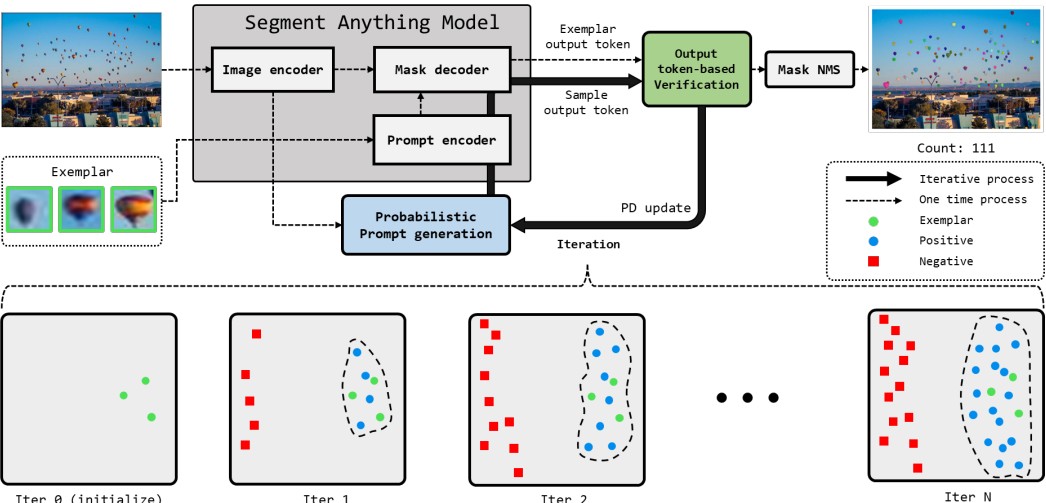

Figure 2: Overview diagram of the proposed method (top) and example of iterative process (bottom). PD means probability distribution.

tively correlated with the exemplar as a non-relevant baseline, thereby isolating features of interest. Subsequently, to accentuate high-similarity regions corresponding to the foreground, we apply temperature scaling with a parameter $\tau$. Finally, a softmax function transforms these scaled scores into a normalized probability distribution across the image. This procedure is formulated as follows:

$$
\begin{aligned}
\mathbf{F}^I &= f(\mathbf{I}) \\
\mathbf{M} &= g(\mathbf{F}^I, \boldsymbol{e}) \\
\mathbf{M}^R &= \mathrm{resize}(\mathbf{M}, (64, 64, 1)) \\
\mathbf{F}^E &= \mathrm{nonzero}(\mathbf{F}^I \bigcirc \mathbf{M}^R) \\
\boldsymbol{S} &= \cos(\mathbf{F}^I, \mathrm{mean}(\mathbf{F}^E)) \\
P(\mathbf{I}; \boldsymbol{e}) &= \mathrm{softmax}(\frac{\mathrm{ReLU}(\boldsymbol{S})}{\tau}),
\end{aligned}
\tag{1}
$$

where $\mathbf{I} \in \mathbb{R}^{1024 \times 1024 \times 3}$ is an input image, $f$ is image encoder of SAM, and $\mathbf{F}^I \in \mathbb{R}^{64 \times 64 \times 256}$ is image embedding. $\boldsymbol{e}$ is a given exemplar prompt, $g$ is mask decoder of SAM, $\mathbf{M} \in \mathbb{R}^{256 \times 256 \times 1}$ is an output mask of SAM. $\mathbf{M}^R \in \mathbb{R}^{64 \times 64 \times 1}$ is resized mask and, $\bigcirc$ is Hadamard product. $\mathbf{F}^E \in \mathbb{R}^{N \times 256}$ is image embedding of mask area, and $N$ is the number of mask pixels. $\boldsymbol{S}$ is a similarity matrix and $P(\mathbf{I}; \boldsymbol{e})$ is final probability distribution for prompt generation.

Based on the generated probability distribution, prompts are sampled from a multinomial distribution. This calculation ensures prompts are placed in positions semantically similar to the example. High values are assigned to similar areas, while low values are assigned to dissimilar areas. Initial example output tokens are sparsely distributed. Due to inherent positional information of output tokens, objects that should be verified positively but are positionally distant from the initial output tokens may fail verification. To address this issue, we designed an iterative algorithm that starts from nearby positive objects and gradually expands the search range to more distant objects. In the $m$th iteration, we sample $k$ prompts from the multinomial distribution. If an output token is validated

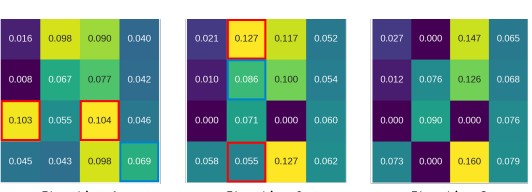

Figure 3: Example of probability distribution update process. The red boxes are verified as positive and change to 0 in the next iteration, while the blue boxes are verified as negative and their values decrease in the next iteration.

as positive, we add that output tokens and its mask to the exemplar list to use as new verification criteria in the next iteration.

To prevent multiple prompts from targeting the same object, the probability values of the mask regions of verified output tokens are decreased. As shown in Figure 3, when classified as positive, the probability distribution at the corresponding coordinate is set to 0, and when classified as negative, the value of the probability distribution at that mask region is decreased. This is designed to reduce redundant searches for negative coordinates with high probability distribution values. By changing the probability distribution, prompts are efficiently dispersed across objects instead of clustering around a single target. As the search expands to undiscovered objects, this adaptive method achieves a balance between computational efficiency and accuracy. Through the iterative structure, we set a maximum iteration limit to balance thoroughness and efficiency. Each iteration follows a priority order, starting from areas with the highest probability values and gradually moving to areas with lower probabilities. This ensures systematic coverage while maintaining computational tractability.

### 3.2 OUTPUT TOKEN-BASED VERIFICATION

We leverage the output tokens from SAM's decoder as the primary verification criteria and TS-SS (Triangle Similarity-Sector Similarity) as the verification metric for object instance counting(Heidarian & Dinneen, 2016). This framework effectively transforms object counting into a clustering problem solved via vector similarity. We accurately identify objects of the same category by measuring the similarity between the token representation of exemplars and those of all candidate objects in a query image.

Output tokens are generated through a series of bidirectional attention operations within the transformer-based decoder. Initially, self-attention among the output tokens and prompt embeddings establishes spatial grounding. Subsequently, in token-to-image cross-attention, these tokens aggregate relevant visual features such as color, texture, and boundaries from the image embedding based on this positional context. Conversely, image-to-token cross-attention update the image embedding itself by injecting prompt information back into it, creating a prompt-aligned representation. This iterative process culminates in a final token-to-image attention step, where the fully refined tokens attend to the updated image embedding to capture the precise details required for mask prediction.

A key challenge arises from the nature of the output tokens generated through the process described above, which jointly encode semantic(angular) and positional(magnitudinous) information. It is precisely this entanglement that renders the similarity metrics commonly employed in counting tasks inadequate for reliably verifying these dual-natured tokens. The dot product creates ambiguity by conflating vector magnitude with angular alignment. Cosine similarity discards all vector magnitude information to focus solely on direction. Euclidean distance operates solely on proximity in the embedding space, making it agnostic to semantic direction. To address these limitations, we employ TS-SS, a robust similarity metric designed to evaluate the geometric characteristics of output tokens.

This metric consists of the TS that measures similarity based on the area of the triangle and the SS that measures similarity based on the sector area considering both angle and magnitude between vectors. These are computed as

$$\text{TS}(\boldsymbol{a}, \boldsymbol{b}) = \frac{|\boldsymbol{a}| * |\boldsymbol{b}| * \sin(\boldsymbol{\theta})}{2}, \quad \text{SS}(\boldsymbol{a}, \boldsymbol{b}) = \pi * (\text{ED}(\boldsymbol{a}, \boldsymbol{b}) + \text{MD}(\boldsymbol{a}, \boldsymbol{b}))^2 * \frac{\boldsymbol{\theta}}{360°}, \quad (2)$$

where $\boldsymbol{\theta}$ is an angle of the two vectors, ED is euclidean distance, and MD is manhattan distance.

$$\text{TS-SS}(\boldsymbol{a}, \boldsymbol{b}) = \text{TS}(\boldsymbol{a}, \boldsymbol{b}) * \text{SS}(\boldsymbol{a}, \boldsymbol{b}), \quad (3)$$

This metric is calculated as the product of two core components, creating a non-linear relationship that offers more sophisticated analysis. Each component effectively handles the dual nature of the output token space. Furthermore, to complement and maximize the efficacy of our output token-based verification, we introduce in Section 3.1 a probabilistic prompt generation stage that ensures comprehensive spatial coverage, enabling the effective exploitation of encoded positional information. The effectiveness of the TS-SS metric against standard metrics is demonstrated through extensive experiments, which are further detailed in the ablation study.

# 4 EXPERIMENT

## 4.1 EXPERIMENTAL SETUP

We evaluated our method on two widely used object counting benchmark datasets: FSC-147 and CARPK. The FSC-147 dataset is a comprehensive collection containing 147 different object categories with a diverse range of counting scenarios. The CARPK dataset specifically focuses on aerial views of parking lots, presenting unique challenges in vehicle counting from elevated perspectives. This specialized dataset offers an opportunity to evaluate counting performance in real-world scenarios where objects maintain relatively consistent sizes but appear under varying lighting conditions, orientations, and degrees of occlusion. For CARPK, following the testing procedure, we randomly selected 12 objects from the training set to use as exemplars.

We used standard metrics to evaluate counting performance: Mean Absolute Error (MAE), Root Mean Square Error (RMSE), Normalized Absolute Error (NAE) and Squared Relative Error (SRE). In particular, $\text{MAE} = \frac{1}{n}\sum_{i=1}^{n}|y_i - \hat{y}_i|$, $\text{RMSE} = \sqrt{\frac{1}{n}\sum_{i=1}^{n}(y_i - \hat{y}_i)^2}$, $\text{NAE} = \frac{1}{n}\sum_{i=1}^{n}\frac{|y_i - \hat{y}_i|}{y_i}$, $\text{SRE} = \sqrt{\frac{1}{n}\sum_{i=1}^{n}\frac{(y_i - \hat{y}_i)^2}{y_i}}$.

**Implementation detail.** We implemented the system using PyTorch and the official SAM. For point prompt generation, we configured the SAM predictor with a t-scale factor of 0.066. Based on validation experiments, we set the token verification threshold to 300. All experiments were conducted on a single NVIDIA A100 GPU with 40GB memory, achieving an average processing time of 1.69 seconds per image on the FSC-147 dataset. To handle redundancy, we applied mask-based non-maximum suppression (NMS) to remove overlapping masks with high intersection over union (IoU) values. Since we applied NMS at the segmentation mask level rather than the bounding box level, we could overlapping masks more precisely.

Table 1: Performance comparison on FSC-147 and CARPK datasets. Shows MAE, RMSE, NAE, SRE metrics for training-based and training-free methods.

| Method | Train | FSC-147 | | | | CARPK | | | |
|---|---|---|---|---|---|---|---|---|---|
| | | **MAE** | **RMSE** | **NAE** | **SRE** | **MAE** | **RMSE** | **NAE** | **SRE** |
| FamNet | Yes | 22.08 | 99.54 | 0.44 | 6.45 | 18.19 | 33.66 | - | - |
| CFOCNet+ | Yes | 22.10 | 112.71 | - | - | - | - | - | - |
| C-DETR | Yes | 16.79 | 123.56 | 0.19 | 5.23 | - | - | - | - |
| BMNet+ | Yes | 14.62 | 91.83 | 0.25 | 2.74 | 5.76 | 7.83 | - | - |
| SAFECount | Yes | 14.32 | 85.54 | - | - | 5.33 | 7.04 | - | - |
| PseCo | Yes | 13.05 | 112.86 | - | - | - | - | - | - |
| LOCA | Yes | 10.79 | 56.97 | - | - | 9.97 | 12.51 | - | - |
| **SAM + $\alpha$** | | | | | | | | | |
| OmniCount | No | 18.63 | 112.98 | 0.14 | 2.99 | 9.92 | 12.15 | 0.23 | 2.11 |
| TFCounter | No | 18.56 | 130.59 | - | - | 9.71 | 12.44 | - | - |
| TFCAC | No | 12.26 | 56.33 | - | - | 4.39 | 5.70 | - | - |
| **SAM only** | | | | | | | | | |
| SAM | No | 42.48 | 137.50 | 1.14 | 8.13 | 16.97 | 20.57 | 0.70 | 5.30 |
| C-Anything | No | 27.97 | 131.24 | - | - | - | - | - | - |
| TFOC | No | 19.95 | 132.16 | 0.29 | 3.80 | 10.97 | 14.24 | 0.48 | 3.70 |
| Ours | No | 16.25 | 133.11 | 0.15 | 3.23 | 4.68 | 6.13 | 0.08 | 0.74 |

## 4.2 QUANTITATIVE MEASUREMENT

In the few-shot counting scenario, we compared our proposed method with training-based density map-based object counting methods such as FamNet, CFOCNet, BMNet, SAFECount, PseCo, LOCA, and the detection-based object counting method Counting-DETR. We also compared with training-free counting methods including Count-Anything, TFOC, OmniCount, TFCounter, and TF-CAC. Including our proposed method, these methods used SAM as their base model. For detailed comparison, we categorized and analyzed methods as either SAM+$\alpha$ or SAM only based on whether

they use additional foundation models. Through this distinction, we can more clearly evaluate the efficiency and effectiveness of each approach. Experimental results are summarized in Table 1.

Table 2: Comparison of one-shot object counting methods on FSC-147 dataset.

| Methods | Training | FSC-147 | |
|---|---|---|---|
| | | MAE | RMSE |
| CFOCNet | Yes | 28,60 | 123.96 |
| FamNet | Yes | 26.76 | 110.95 |
| BMNet+ | Yes | 16.89 | 96.65 |
| LOCA | Yes | 12.53 | 75.32 |
| **SAM + $\alpha$** | | | |
| TFCAC | No | 15.15 | 69.16 |
| **SAM only** | | | |
| C-Anything | No | 33.53 | 142.28 |
| TFOC | No | 22.96 | 137.27 |
| Ours | No | 18.57 | 135.86 |

On the FSC-147 dataset, our proposed method achieved the best performance among training-free SAM only methods with an MAE of 16.25. This is superior to the training-based method Counting-DETR's 16.79. Compared with SAM+$\alpha$ methods, our method shows improved performance over OmniCount and TFCounter. TFCAC achieved performance improvements by using an additional image encoder of DINO(Caron et al., 2021; Oquab et al., 2023), but this comes with the disadvantage of lower computational efficiency. In contrast, our method achieves high accuracy using only SAM without additional models, maintaining a good balance between computational efficiency and performance.

On the CARPK dataset, we recorded the MAE of 4.68 and RMSE of 6.13, outperforming all training-based methods and training-free methods except TFCAC. Although our proposed method is inferior in performance to the state-of-the-art TFCAC, it features an inference time that is more than twice as fast. This is attributed to TFCAC's reliance on an additional model and a greater number of prompts. The consistency of viewpoint and scenario is particularly advantageous for our method, because SAM excels in segmenting objects with clear boundaries and predictable appearances. This explains why our method achieves high performance metrics compared to both training-based and training-free methods in this case.

**One-shot counting.** As shown in Table 2, our experimental results demonstrate competitive performance against both training-based and training-free one-shot methods. In the FSC-147 dataset, our method achieves an MAE of 18.57, comparable to training-based methods such as BMNet+, and outperforms some training-based methods like CFOCNet. Although our RMSE of 135.86 is higher than some training-based methods, the results are promising. Among the training-based methods, LOCA shows the strongest performance in terms of MAE.

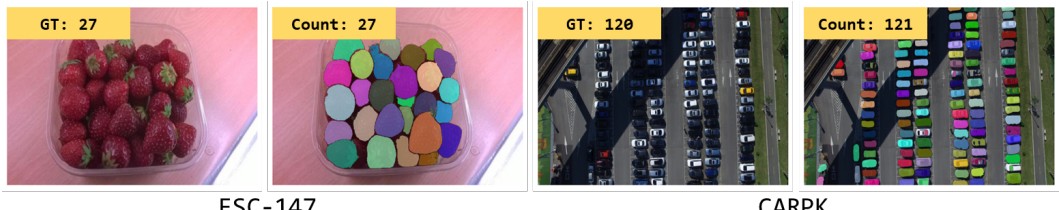

FSC-147          CARPK

Figure 4: Visualization of example results on FSC-147 and CARPK.

### 4.3 QUALITATIVE ANALYSIS

Figure 4 shows example results for representative scenarios in the FSC-147 and CARPK benchmarks. Additional visualization materials are detailed in the Appendix B. The proposed method works effectively when objects are clearly distinguishable and maintains stable performance even as the number of objects increases. Particularly in the CARPK benchmark, where our method accu-

rately identifies and counts vehicles across various parking lot layouts and densities, our predicted counts closely approximate the ground truth values. The consistent performance across these examples demonstrates the robustness of our method in handling diverse real-world scenarios, including various lighting conditions, vehicle orientations, and parking space arrangements.

Table 3: Ablation study on different verification metric for distinguishing output tokens on the FSC-147 dataset.

| Metric | FSC-147 | |
| --- | --- | --- |
| | MAE | RMSE |
| Dot product | 19.58 | 134.51 |
| Cosine Sim | 16.48 | 133.94 |
| L2 distance | 16.53 | 133.34 |
| TS-SS Sim | 16.25 | 133.11 |

Table 4: Ablation study results demonstrate the impact of iteration count and the number of prompts generated per iteration.

| Iteration | Prompts | FSC-147 | | Mean prompts |
| --- | --- | --- | --- | --- |
| | | MAE | RMSE | |
| 8 | 32 | 29.10 | 142.16 | 252.96 |
| | 64 | 22.94 | 138.17 | 499.57 |
| | 128 | 18.90 | 135.41 | 987.53 |
| 16 | 32 | 21.74 | 137.33 | 416.86 |
| | 64 | 18.13 | 134.87 | 767.73 |
| | 128 | 16.25 | 133.11 | 1438.22 |
| 32 | 32 | 18.47 | 134.95 | 503.15 |
| | 64 | 17.00 | 133.88 | 871.20 |
| | 128 | 16.65 | 132.69 | 1596.45 |

## 4.4 ABLATION STUDY

**Verification metric.** The performance hierarchy in 3 offers important clues about the geometric structure of the output token space. It reveals that although both semantic and positional information are useful, the former holds a slight advantage in this task. Most notably, the near-parity in performance between cosine similarity (16.48) and L2 distance (16.53) strongly implies that non-hybrid metrics have reached a performance plateau. Breaking through this barrier necessitates a new hybrid metric that, instead of choosing one information type, effectively integrates both. This establishes a clear and logical motivation for adopting a geometric approach like TS-SS.

**Iterative sampling.** To maximize efficiency, we implemented an early stopping mechanism with global termination criterion. The system terminates the iterative prompt generation and verification process when it distinguish that all target objects in the query image have been found. This mechanism ensures comprehensive object detection while minimizing computational overhead. We check if the IoU between the mask union from the previous iteration and the mask union from the current iteration exceeds 0.95, and set the patience value to 5 to terminate unnecessary iterations early.

Our ablation study evaluated the impact of iteration count and prompts per iteration on performance by applying early stopping mechanism. As shown in Table 4, we can observe that MAE and RMSE performance consistently improves as the number of prompts in each iteration increases to 32, 64, and 128. This proves that discrimination using output token similarity in the verification stage works correctly. As iterations and prompts increase, the average number of prompts across the entire test set also increases. This presents practitioners with an important balance point between counting accuracy and computational efficiency when implementing our method.

**Verification criteria.** We performed these experiments to prove that our proposed method, which counts objects with high similarity to exemplars using output tokens, outperforms existing research that counts objects with high similarity to exemplars in the image embedding space. As explained

Table 5: Performance comparison between exemplar vectors and generated prompt vectors in both image embedding space and output token space.

| Methods | Vector Space | FSC-147 | |
| --- | --- | --- | --- |
| | | MAE | RMSE |
| Ours | Output token | 16.25 | 133.11 |
| | Image embedding | 16.94 | 133.71 |

in Section 3, output tokens extract semantic information from prompted position information and then integrate it with position information. This provides a means to clearly distinguish similarities between objects, compared to the image embedding space where visually similar objects may share similar features and identification can be ambiguous. As shown in Table 5, the performance metrics in the output token space are high, proving our claim is accurate.

As shown in Figure 5, when we sample the image embeddings and output tokens at corresponding coordinates for the same point prompts as the input image and plot them with 2D PCA, we observe the following results. For the top image, in the case of image embeddings, the positive objects are distributed separately. The bottom image fails to correctly classify positive and negative objects. In contrast, the output tokens clearly distinguish between positive and negative objects.

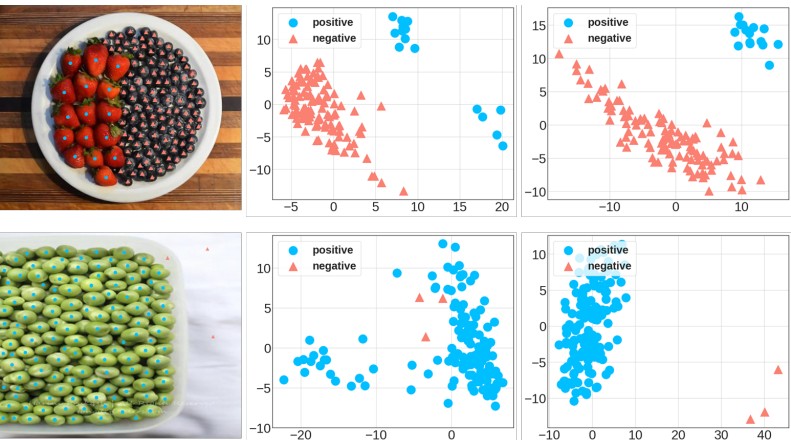

Figure 5: Visualization of feature distribution in different spaces using PCA. The images in the first column show positive and negative point prompts. The second column shows how features from positive and negative prompts overlap and are not clearly separated in the image embedding space. The third column shows how positive and negative prompts form distinct and clearly separable clusters in the output token space.

## 5 DISCUSSION

Our experiments reveal several key insights and limitations of the proposed methodology. While our method demonstrates strong performance across diverse categories without training, it shows performance degradation in scenarios with dense distributions of very small instances. This occurs because SAM fails to distinguish correct boundaries during mask generation, causing the token-based verification stage to struggle in maintaining consistent discrimination criteria. These limitations suggest potential areas for future research to enhance the robustness of training-free counting methods. Our method, composed solely of SAM without additional models, demonstrates superior computational efficiency compared to state of the art. This limitation affects scalability for real-time applications and large datasets.

## 6 CONCLUSION

In this paper, we present a novel training-free object counting framework that combines probabilistic prompt generation and output token-based verification while utilizing SAM's original structure. We achieve competitive performance across object counting benchmarks, particularly showing superior performance compared to training-based methods on the CARPK dataset. Experimental results validate the effectiveness of our method, demonstrating performance equal to or better than existing methods while maintaining the advantages of training-free deployment. We have demonstrated the effectiveness of each proposed stage through ablation studies. Additionally, studying how to optimize computational efficiency for large-scale applications while maintaining accuracy remains an important challenge to be addressed.

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

## A    THE USE OF LLMS

This paper utilized LLM assistance in part for sentence structure improvement and vocabulary correction during the writing process.

## B    VISUALIZATION OF EXAMPLE RESULTS

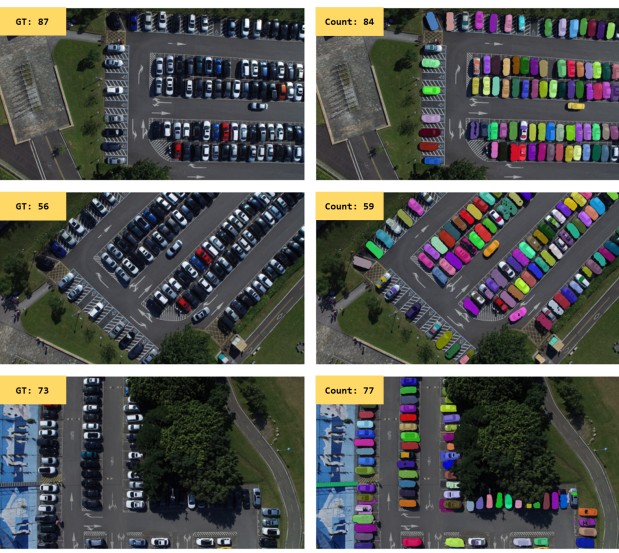

Figure 6: Qualitative results on the CARPK dataset. Left column shows original aerial parking lot images with ground truth (GT) counts, while right column displays our method's predicted instance masks. The results demonstrate accurate vehicle counting across various parking lot configurations and densities.

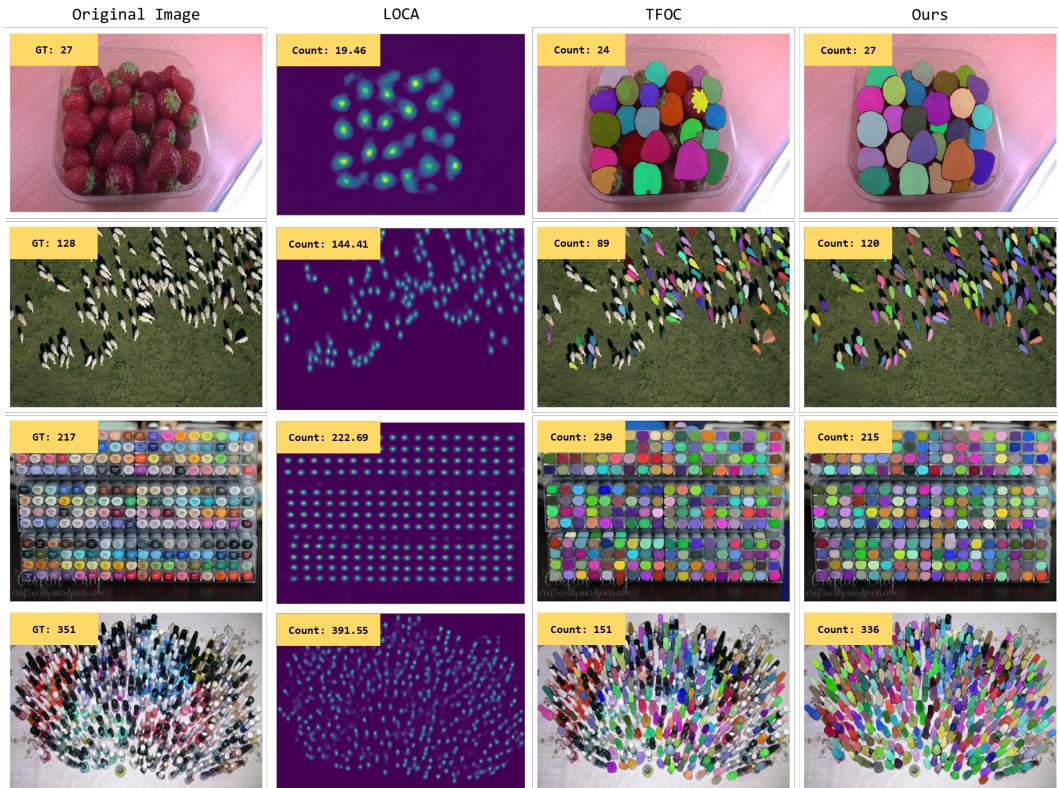

Figure 7: Qualitative results on the FSC-147 dataset. Left column shows original images with ground truth (GT) counts. The second column displays predicted density distribution and counts of LOCA. The third and fourth columns show the predicted instance segmentation masks and counts for TFOC and our proposed method, respectively.

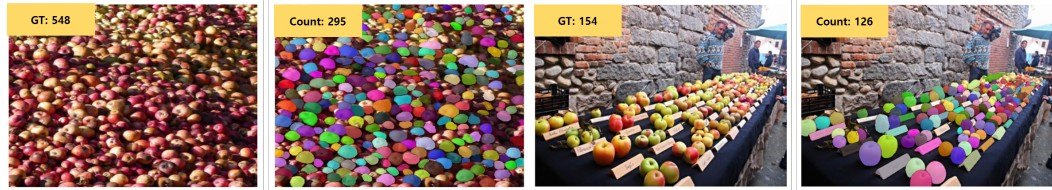

Figure 8: Qualitative result of failure case for the FSC-147 dataset. The first and third column show original images and ground truth of counts. The second and fourth column show predicted masks and counts.

Figure 6 demonstrates the effectiveness of our method on the CARPK dataset, which consists of aerial images of parking lots. The visualizations show pairs of original images and our model's instance segmentation results. Our method accurately identifies and counts vehicles across various parking lot layouts and densities, with predicted counts close to ground truth values. The consistent performance across these examples demonstrates the robustness of our method in handling diverse real-world scenarios, including various lighting conditions, vehicle orientations, and parking space arrangements.

Figure 7 shows example results for various scenarios in the FSC-147 benchmark. We qualitatively compared our method with LOCA, the state of the art among training-based methods, and TFOC, a representative case of training-free methods. The proposed method works effectively when instances are clearly distinguishable (first row) and maintains stable performance even as the number of objects increases. In all four cases, we can confirm that our proposed output token-based counting

method distinguishes instances more accurately than TFOC, which uses a similarity measurement method based on image embeddings.

Figure 8 Analyzing failure cases reveals patterns where this method struggles. Generally, these cases occur in images with dense and overlapping objects, and in cases with objects of abnormal sizes that differ significantly from the exemplar. These limitations suggest the need for a more robust approach to handle exceptional cases.

