# OpenReview forum: "TokenCount: A Training-Free Framework for Object Counting by Interpreting Output Tokens"
_ICLR.cc/2026/Conference — Submitted to ICLR 2026_

### Official Review · Reviewer_tXFk · 2025-10-26

**Soundness:** 2
**Presentation:** 1
**Contribution:** 2
**Rating:** 2
**Confidence:** 4

**Summary:**

This paper introduces a method for class-agnostic object counting that fully exploits SAM without any additional training or auxiliary models. It combines probabilistic prompt generation, which adaptively explores image regions based on SAM’s embeddings, with output token-based verification, which directly interprets SAM’s decoder tokens using a geometric similarity metric (TS-SS) to distinguish true object instances. The method achieves competitive or superior accuracy to both training-based and training-free approaches on FSC-147 and CARPK datasets while maintaining computational efficiency.

**Strengths:**

(1) This paper introduces a training-free framework that leverages SAM’s architecture without any fine-tuning or auxiliary encoders.

(2) It achieves strong results, outperforming prior training-free and some supervised methods, with comprehensive ablation and qualitative analyses validating each component’s effectiveness.

**Weaknesses:**

(1) The probabilistic prompt generation module iteratively classifies masks as positive or negative and generates new prompts accordingly. However, it is unclear how these classifications are determined. For example, in Figure 3, during the second iteration, masks with probabilities 0.127 and 0.055 are marked as positive, whereas the mask with a much higher probability of 0.86 is marked as negative, and the one with 0.071 is not selected at all. Could the authors clarify the criteria or mechanism behind this selection process?

(2) The iterative probabilistic prompt generation process likely increases time complexity. However, despite the method’s emphasis on efficiency and its training-free design, computational scalability and real-time applicability have not been evaluated on large-scale or high-resolution datasets.

(3) It is unclear how the TS-SS metric operates in practice. While it is defined as a similarity measure between two vectors, the paper does not clearly explain how this metric facilitates clustering of similar objects, as mentioned in lines 234–235.

**Questions:**

(1) Since TS-SS combines angular and magnitude components, how is it normalized to handle scale variations in token embeddings? Is there empirical or theoretical justification that TS-SS induces a clustering structure suitable for object instance separation? What is the motivation on designing TS-SS metric?

(2) What is the computational complexity (in terms of prompts or tokens) per iteration, and how does it scale with image size or object density? Have the authors benchmarked inference time or memory usage across varying resolutions (e.g., 512×512 vs. 2048×2048)?

---

### Official Review · Reviewer_vU7v · 2025-10-27

**Soundness:** 2
**Presentation:** 2
**Contribution:** 2
**Rating:** 4
**Confidence:** 4

**Summary:**

This paper introduces a training-free, class-agnostic object counting method. Built upon Segment Anything Model (SAM), the proposed pipeline consists of two stages, namely probabilistic prompt generation and output token-based verification. The former samples point prompts according to the probability distribution derived from SAM’s image encoder, while the latter identifies positive and negative instances by analyzing the output tokens of SAM. These two stages operate iteratively until a predefined criterion is met. Evaluations on two benchmarks show that the proposed method performs favorably against training-based and training-free counting methods.

**Strengths:**

* The paper demonstrates how SAM can be repurposed as a training-free, class-agnostic object counting model, highlighting SAM’s potential for specialized downstream tasks without additional training.
* The proposed method comprises two modules, including probabilistic prompt generation and output token-based verification. These modules iteratively generate point prompts and predict objects, achieving competitive counting performance.

**Weaknesses:**

* The significance and benefits of the proposed training-free method require further clarification. Admittedly, training-free SAM-based counting was indeed a promising direction in 2023. However, a large number of advanced methods [1,2,3] have been proposed in 2024, which substantially outperform training-free methods. The paper would benefit from a more nuanced discussion of where the proposed method stands relative to recent methods and what specific advantages it offers.
* Missing analysis on computational efficiency. As the authors claim that the proposed method could enhance computational efficiency by reducing redundant prompt generation, it would be necessary to analyze the computational cost of each module and compare with vanilla SAM baselines (e.g., inference with a fixed number of point prompts).
* The necessity of the proposed output token vector space remain questionable. As shown in Table 5, using the original image embedding already achieves similar performance compared to the output token-based variant (16.94 MAE vs. 16.25 MAE). In addition, visualizations in Figure 5 do not necessarily support that the output token space is better than image embedding, as the positive and negative objects are still separable under image embedding setting.
* The proposed similarity metric, dubbed TS-SS Sim, does not show superiority over existing metrics such as cosine similarity and L2 distance. As shown in Table 3, replacing cosine similarity (16.48 MAE) with TS-SS Sim yields negligible improvement (16.25 MAE). This raises concerns regarding the necessity of TS-SS Sim.
* While zeroing out negative values in the similar map helps reduce redundant background prompt sampling, it is possible that those discarded regions contain true positive objects, particularly in congested and challenging scenarios. It would be beneficial to analyze when does the probabilistic distribution fail to capture actual objects.

**Reference**

[1] CountGD: Multi-Modal Open-World Counting. NeurIPS 2024.

[2] A Novel Unified Architecture for Low-Shot Counting by Detection and Segmentation. NeurIPS 2024.

[3] DAVE – A Detect-and-Verify Paradigm for Low-Shot Counting. CVPR 2024.

**Questions:**

Please refer to the weakness section.

---

### Official Review · Reviewer_9GU5 · 2025-10-30

**Soundness:** 3
**Presentation:** 3
**Contribution:** 2
**Rating:** 4
**Confidence:** 5

**Summary:**

This paper introduces TokenCount, a training-free, class-agnostic counting (CAC) framework that reuses Segment Anything (SAM) to bypass the need for additional training or fine tuning. The proposed method leverages SAM decoder tokens for similarity computation to improve interpretability in CAC relative to methods that rely on image-level embeddings, and it introduces the TS-SS geometric metric as the similarity metric for object instance count. In terms of architecture, TokenCount couples probabilistic prompt generation with output token verification. The former generates prompts by iteratively sampling from probability distributions derived from image embeddings of SAM, and the latter uses decoder output tokens of SAM to distinguish positive from negative instances. On FSC 147 and CARPK, TokenCount achieves SOTA performance among methods that rely solely on SAM for counting.

**Strengths:**

- **Profound insights**: The paper provides compelling evidence where decoder tokens provide a more effective, instance-aware representation for counting than image embeddings. It recasts class-agnostic counting as training-free exemplar matching on SAM decoder tokens and combines this novel representation with probabilistic prompt generation (PPG) with the TS-SS geometric similarity metric.
- **Robust performance**: The method demonstrates competitive accuracy on FSC-147 and CARPK among training-free baselines that rely solely on SAM. These gains are consistent across shot regimes and target densities, with the effectiveness further validated by ablation studies.
- **Simple pipeline**: The method uses SAM in an end-to-end manner and does not require external encoders, detectors, or fine-tuning, rendering the algorithmic steps concise and reproducible.

**Weaknesses:**

-	**Efficiency concerns**: The probabilistic prompt generation iterations require repeated decoder computations, leading to substantial computational overhead and efficiency degradation.
-	**Lack of convergence guarantees**: The probabilistic prompt generation iterations lack formal convergence guarantees and do not ensure separability between positive and negative samples in the feature space. This limitation precludes establishing a lower bound on the performance.
-	**Unclear method description**: The section on output token verification does not explain how the final count is produced. While the similarity metrics are well discussed, the methodological pipeline remains incomplete. The mechanism and parameters for Non-Maximum Suppression (NMS) or clustering are missing.
-	**Inadequate discussion and analysis**: The paper emphasizes token-based counting over embedding-based counting. A rigorous analysis of failure cases and a sound examination, at least an empirical one, of the iteration convergence are essential. However, the Discussion section only briefly addresses these issues, and a deeper exploration would enhance the contribution of this paper.

**Questions:**

-	If TokenCount treats a target as a single token, how does it handle larger targets that span multiple tokens? What failure modes are likely to occur due to the aggregation threshold, and how can they be mitigated?
-	Regarding Table 4, several questions arise about the definition and use of prompts. The paper does not clearly describe the ablation setup. What does the reported number of prompts denote? For example, if the first row uses 8 iterations with 32 prompts per iteration, the theoretical upper bound is 256 prompts. Why does the observed average number of prompts already approach this bound?

**Details Of Ethics Concerns:**

N.A.

---

### Official Review · Reviewer_WjNt · 2025-10-31

**Soundness:** 2
**Presentation:** 2
**Contribution:** 2
**Rating:** 4
**Confidence:** 5

**Summary:**

This paper introduces TokenCount, a training-free, class-agnostic object counting framework built entirely upon the native architecture of the Segment Anything Model (SAM). The core contributions are twofold: (1) a probabilistic prompt generation stage designed to efficiently sample potential object regions within an image, and (2) an output token-based verification stage that identifies and distinguishes target instances by directly measuring the similarity of SAM's decoder output tokens.

**Strengths:**

The authors conduct experiments on the FSC-147 and CARPK benchmarks, demonstrating that their method achieves strong performance among training-free approaches, even outperforming some training-based methods on certain metrics, particularly when using only SAM without auxiliary models. The idea of leveraging output tokens for direct verification to circumvent the ambiguity of similarity metrics in the image embedding space is clever.

**Weaknesses:**

The paper presents probabilistic prompt generation as a key contribution to improve efficiency over grid-based or superpixel sampling. However, the core of this stage is the computation of a cosine similarity map between the exemplar and query image embeddings, followed by sampling from it. This idea is quite common in exemplar-based methods and is closely related to the principle of attention mechanisms. While the proposed iterative update strategy for the probability distribution is a nice implementation detail, the fundamental concept lacks sufficient novelty. The paper would better highlight its unique contribution by providing a more thorough comparison with other adaptive or attention-based sampling strategies.


The paper acknowledges performance degradation in scenarios with "dense distributions of very small instances," attributing it to SAM's failure to generate clean segmentation masks, which in turn hampers the token verification stage . This is a critical limitation for a general-purpose counting method. While the discussion section mentions this, it lacks a deeper analysis. For instance, how does the probabilistic prompt generation strategy behave in these dense regions? Does it over-sample or under-sample? How exactly does token verification fail when masks heavily overlap? Providing more failure cases like those in Figure 8 and dissecting them with respect to each module of the method would help to more clearly define the method's application boundaries .


The paper describes an iterative process where newly identified positive samples are added to the exemplar list, and the probability map is updated to suppress already-verified regions . This process is crucial to the final performance, yet its dynamics are not fully explored.

**Questions:**

see weakness

---

### Meta-Review · Area_Chair_64dm · 2026-01-06

**Summary:**

In the initial phase, all reviewers gave negative scores (4, 4, 4, 2). They raised concerns from different aspects.

Reviewer WjNt has concerns on the novelty, limitation of counting small objects, and lack of study on the iterative process.
Reviewer 9GU5 has concerns on the efficiency, lack of convergence guarantees, unclear description, and insufficient analysis.
Reviewer vU7v has concerns on the efficiency, the importance of training-free counting approach, and the necessity of TS-SS.
Reviewer tXFk has concerns on the efficiency, and lack of details regarding the method.

**Reviewer Concerns:**

Since no rebuttal is provided, all concerns remain unsolved.

**Reviewer Scores:**

Since no rebuttal is provided, the scores will remain unchanged.

---

### Decision · Program_Chairs · 2026-01-26

Reject